# Postoperative Trends of Serum C-Reactive Protein Levels after Primary Shoulder Arthroplasty—Normal Trajectory and Influencing Factors

**DOI:** 10.3390/jcm9123893

**Published:** 2020-11-30

**Authors:** Sebastian Klingebiel, Jan Christoph Theil, Georg Gosheger, Kristian Nikolaus Schneider, Maximilian Timme, Dominik Schorn, Dennis Liem, Carolin Rickert

**Affiliations:** 1Department of Orthopedics and Tumororthopedics, University Hospital Muenster, 48149 Münster, Germany; Christoph.Theil@ukmuenster.de (J.C.T.); Georg.Gosheger@ukmuenster.de (G.G.); kristian.schneider@ukmuenster.de (K.N.S.); Carolin.Rickert@ukmuenster.de (C.R.); 2Department of Forensic Medicine, Institute for Legal Medicine, University Hospital Muenster, 48149 Münster, Germany; M.timme@uni-muenster.de; 3Department for Shoulder and Elbow Surgery, Paracelsus Clinic Bremen, 28329 Bremen, Germany; schorn.ortho@gmail.com; 4Sporthopaedicum Berlin, Medical Practice, 10627 Berlin, Germany; Liem@sporthopaedicum.de

**Keywords:** C-reactive protein, trajectory, trend, normal curve, postoperative CRP course, laboratory diagnostics, total shoulder arthroplasty, reverse shoulder arthroplasty, TSA, RSA, glenohumeral joint replacement, periprosthetic joint infection, PJI, hemiarthroplasty

## Abstract

Background—Postoperative serum C-reactive protein (CRP) is an important diagnostic parameter for systemic inflammation and reflects surgical trauma. While trends and normal trajectories after total knee (TKA) or hip arthroplasty (THA) are established, there is no reference standard for shoulder arthroplasty (SA). Therefore, the aim of this study was to research CRP trends and influencing factors following SA. Methods—This retrospective study analyzed postoperative serum CRP levels and trajectories in 280 patients following SA. Influence of prosthesis design, sex, operating time, BMI, and humeral augmentation with bone cement were analyzed using descriptive statistics and (non-) parametric testing. Results—There is a CRP trend with a peak on day two or three, with a subsequent decrease until day seven. Reverse and stemmed prostheses show a statistically higher CRP peak than stemless prostheses or hemiarthroplasties (HA). There was no influence of gender, body mass index (BMI), operating time, or bone cement. Conclusion—The presented findings may contribute to a better understanding of the postoperative CRP course after SA. The results of this retrospective study should be validated by a prospective study design in the future.

## 1. Introduction

Laboratory testing, and particularly blood sampling, is an indispensable tool for surveillance of perioperative blood loss and inflammation parameters after orthopedic surgeries, especially following major interventions such as joint arthroplasty [1]. In this context, C-reactive protein is an established parameter. This acute phase protein is involved in the early systemic inflammation response [2]. C-reactive protein (CRP) is synthesised in hepatocytes—for instance, during infectious processes and following tissue trauma or neoplasia. [3]. Local tissue damage leads to a cytokine cascade, with release of interleukin-6 and other essential mediators triggering a rapid increase of serum CRP within 12–24 h [4,5]. Knowledge of the standard postoperative course of serum CRP can be helpful in identifying unusual postoperative patterns. Monitoring the trend of serum CRP may therefore indicate a regular postoperative inflammatory pattern or reveal divergent postoperative inflammatory processes such as periprosthetic joint infection (PJI) or other acute diseases with a systemic response [6,7]. Physiological trajectories after various surgical procedures in the lower limb have been reported in the literature. After total knee (TKA) and hip arthroplasty (THA), a peak in serum CRP levels was observed on day two or three, with a subsequent decrease the next few days [8]. Larsson et al. assumed that it is possible to distinguish between early increases due to surgical trauma and delayed increases due to implant-associated early onset infection [9].

Shoulder arthroplasty (SA) is a growing field, but an invasive form of surgical treatment of omarthritis, cuff tear arthropathy, or humeral head necrosis [10]. However, sparse data are available on the trend of CRP levels following SA. Furthermore, the influence of sex, prosthesis design, and further factors have not been identified.

The aim of the present study was to research the trend of CRP trajectory after SA and the analysis of the impact of model of prosthesis, sex, body mass index (BMI), operating time, and the use of polymethylmethacrylate (PMMA) bone cement.

## 2. Patients and Methods

### 2.1. Data Collection

This retrospective study included 280 patients (166 women, 114 men) who underwent elective SA in a single center between November 2010 and October 2019. Data regarding demographic characteristics, diagnosis, treatment, and laboratory parameters were retrieved from the hospital information system (Orbis; AGFA Healthcare, Mortsel, Belgium).

Exclusion criteria were revision surgery, active inflammation or infection, known inflammatory diseases, neoplasia, liver cirrhosis, and postoperative complications within the three-month follow-up. Patients with elevated preoperative baseline CRP level > 1 mg/dL were also excluded (reference < 0.5 mg/dL).

### 2.2. Surgical Treatment

Patients were all treated by an experienced surgeon in SA. All surgeries were performed in a modified beach chair position, with general anesthesia and an additional ultrasound-guided supraclavicular or interscalene plexus catheter. Antibiotic prophylaxis was administered as an intravenous single shot (cefuroxime 1.5 g or clindamycin 600 mg in case of penicillin allergy) 30 min before the incision. Standard postoperative antibiotic prophylaxis was not applied.

All interventions were performed by deltopectoral approach. The cephalic vein was lateralised and preserved whenever possible. The circumflex vessels were coagulated as a standard procedure. Tenodesis of the long bicep tendon was performed with a nonresorbable suture. Joint exposure was achieved by tenotomy of the subscapularis tendon. In case of anatomical joint replacement, the subscapularis tendon was refixed by a double-row technique. In reverse shoulder arthroplasty (RSA), the subscapularis tendon was reattached if the tissue quality and soft tissue tension was appropriate. Augmentation with PMMA bone cement was performed by a closed vacuum system (Copal R + G, Heraeus Medical GmbH, Wehrheim, Germany) to reduce the risk of contamination and to achieve a homogeneous consistency. Patients received a Redon drain (Megro/Ratiomed, Wesel, Germany) for a period of at least 2 days. For a period of 4 weeks following SA, all patients were immobilised in a shoulder brace. First clinical and radiographic follow-up examination in our special orthopedic consultation service was scheduled 3 months after joint replacement.

### 2.3. Laboratory Work

Laboratory tests included preoperative monitoring of the blood cell count and serum CRP was collected for all patients. Further blood collections were performed at least twice after the intervention within the following seven days to identify a trend. Patients were discharged from hospital when the initially raised parameters (CRP, leukocytes) decreased and signs of local or systemic infection were absent. Further, 582 postoperative CRP samples were collected.

Serum CRP was measured using latex-enhanced immunoturbidimetric assay (CRP2) on an Advia 1800 chemistry analyzer (Siemens Healthcare Diagnostics GmbH, Eschborn, Germany). The intra-assay coefficient of variation was determined as 4.7%. The interassay coefficient of variation was determined as 6.2%.

Based on these measurements, statistical analysis was performed to research a postoperative trend and to identify a potential normal trajectory for serum CRP levels following SA. Additionally, the impact of sex- and prosthesis-specific aspects (total shoulder arthroplasty (TSA), RSA, stemless, and stemmed hemiarthroplasty (HA)), as well as BMI, operating time (<90 min, 91–120 min, and >120 min), and the application of bone cement were analyzed.

### 2.4. Statistical Analysis

Statistical analysis of the data was carried out using IBM SPSS Statistics, version 25 (IBM Corporation, Armonk, NY, USA). As the data were non-parametrically distributed, medians are given with 25–75% interquartile ranges (IQR). The Mann-Whitney-U-test was used for nonparametric independent samples of two groups and the Kruskal–Wallis test was used to compare the CRP distribution in more than two groups. Otherwise, means are given with ranges and compared using Student’s *t*-test for independent samples. The significance level was set at the 5% level (*p* < 0.05). Charts and boxplots were produced using Microsoft Excel (Excel 2019, Microsoft Corporation, Redmond, Washington, DC, USA).

The patient data provided a curve and a boxplot for postoperative CRP values. The curve can be assumed to be generally valid, in view of the large database of 280 patients, the corresponding number of blood samples, and the symmetrically distributed values, with the median located largely centrally in the interquartile range 

The study was approved by the university’s ethics committee (Westfälische Wilhelms-Universität Münster (WWU), ref. no. 2020-496-f-S).

## 3. Results

### 3.1. Demographics

A total of 280 patients met the defined criteria undergoing primary SA with absence of postoperative complications within three months after the intervention. Mean age was 66 years (range 28–88 y; females: 69 y, range 30–89 y; males: 61 y, 28–84 y).

Included indications were primary omarthritis in 112 patients (40%), 99 cuff tear arthropathies (35%), secondary omarthritis in 36 patients (12.9%), and 33 necrosis of the humeral head (11.8%). Further, 135 patients (48.2%) underwent RSA, 52 patients (18.6%) received TSA, and 33 patients (11.8%) stemless TSA. Also, 58 patients (20.8%) underwent HA (29 stemless HA (10.4%) and 29 stemmed HA (10.4%)). Two patients (0.7%) received a partial replacement of the humeral head (hemicap). In a total of 52 patients (24.1%), the humeral stem was augmented with PMMA bone cement. Mean operation time was 102.3 min (range 48–380 min). Mean hospitalization after surgery amounted to 6.8 d (range 2–14 d). The average BMI was 29 kg/m^2^ (range 16–46). The BMI was below 30 kg/m^2^ in 176 patients and over 30 kg/m^2^ in 104 (Table 1).

### 3.2. CRP Trend after Shoulder Arthroplasty

In all patients, mean preoperative baseline serum CRP was below 0.5 mg/dL (reference < 0.5 mg/dL) (Table 2). The highest postoperative CRP levels were identified on day three, with a mean of 9.6 mg/dL (IQR 5.3–13.1 mg/dL) (Figure 1). CRP subsequently decreased to 5.0 mg/dL (IQR 2.6–6.2 mg/dL) on day five and to 3.7 mg/dL (IQR 2.0–4.6 mg/dL) on day seven. CRP peak was observed on day two and three in 92% of the patients. In the further 8%, the peak appeared delayed, but with the same trend. Figure 1 and Figure 2 show the postoperative CRP trend with the widest variance on day two and three.

### 3.3. Sex-Specific Postoperative CRP Trend

There were no sex-specific differences in the postoperative CRP trend: For both males and females, the CRP peak was observed on the third day (males: mean 10.0 mg/dL, IQR 5.7–13.4 mg/dL; females: mean 9.3 mg/dL, IQR 5.1–13.1 mg/dL). Up to the seventh postoperative day, the parameter subsequently decreased in both sexes (Figure 3 and Figure 4). A significant difference between, with higher absolute levels in males, was observed on day six (*p* = 0.019).

### 3.4. Prosthesis-Specific Postoperative CRP Trend

Following TSA and HA, the highest levels were reached on day two (stemless TSA: mean 8.2 mg/dl, IQR 3.2–11.0; TSA: mean 8.7 mg/dl, 6.1–13.3; stemless HA: mean 6.9 mg/dl, 4.4–12.9; stemmed HA: mean 9.0 mg/dl IQR 4.5–10.9) (Table 3, Figure 5). After RSA, serum CRP peak was observed on day 3, with a mean of 12.0 mg/dl (IQR 5.6–13.3), which was the highest overall observed mean value (Table 3, Figure 5). The highest absolute serum CRP level followed stemless hemiarthroplasty due to humeral head necrosis with 23.9 mg/dl on day two.

### 3.5. Influence of BMI, Operating Time and Humeral Cement Fixation

No significant differences were found between patients with a BMI less than and higher than 30 kg/m^2^ (Figure 6). Considering the operating time, the described trends in CRP levels were unaffected. In particular, longer surgery duration was not associated with increased CRP levels, except day five, when longest surgery time showed the highest CRP values (*p* = 0.008) (Figure 7). There were no significant differences comparing the trends between cemented and uncemented stems (*p* = 0.136–0.723) (Figure 8).

## 4. Discussion

The present study retrospectively examines the postoperative CRP trends in different types of elective SA. As presented in the results, there is a characteristic trend in over 90% of all patients with the serum CRP peak on the second or third day postoperative followed by a consecutive decrease. The findings of the present study may indicate a postoperative normal trajectory and might help to identify postoperative complications associated with inflammatory response in case of a divergent trend [1]. Being aware of the presented trends might therefore allow early detection of PJI and thus, provide guidance for decision-making on revision surgery or in regard to antibiotic treatment [11,12]. Since only complication-free cases were included in the present study, the results can be seen as an indication that CRP values of 20 mg/dL and higher can be regarded as normal for the considered interventions. From a clinical point of view, it could be suspected that the overall higher mean CRP values following RSA could be related to the more invasive surgical technique due to larger implant sizes. On the other hand, the highest absolute value was registered after implantation of a stemless HA. This could underline that the postoperative CRP trend rather than absolute values should be seen independently of individual surgical techniques and implants. As the operations examined were all performed in a single centre, it can be assumed that the surgical technique for the individual modalities did not differ significantly.

### 4.1. Physiological CRP Kinetics and Clinical Application in Orthopedics

In 1995, Kragsbjerg et al. [13] performed blood collections before, after 6 h, after 12 h and then daily for 6 days in 28 patients undergoing cardiac, orthopedic, or visceral interventions. They reported that serum CRP peak was reached 48–96 h following the intervention. Concentrations were higher than 10 mg/dl for at least 106 h, comparable to patients suffering from sepsis [13]. These postinterventional trends seem analogous to the results of the present study. In 1991, Ellitsgaard et al. [14] analyzed the CRP course after hip fractures in 140 cases [14]. For uncomplicated processes, serum CRP levels presented a clear pattern with a maximum increase on day two and subsequent decrease until day seven. These results are in line with the research of Lim et al. [15], who in 2018, investigated perioperative serologic kinetics of CRP levels after THA for elderly femoral neck fractures [15]. White et al. [8] found CRP peak values of 10.0 mg/dL THA, whereas peak levels after TKA were given as 15.9 mg/dL [16]. Our findings in respect to trends and mean absolute values are comparable to those from White et al. [8] after THA. A reason might be less bony and soft tissue trauma in SA and THA compared to TKA. The idea of using surgical approaches defined by natural muscle gaps are similar in SA and THA, so maybe this fact has an impact on the reduced inflammation reaction in comparison to TKA. Nevertheless, it makes only limited sense to compare absolute values due to the many influencing factors and individual conditions. Niskanen et al. [16] reported a characteristic serologic CRP pattern in patients following elective TKA and THA, with a peak on the second and third postoperative day [16]. Niskanen et al. did not observe any significant differences in the trend relative to the application of bone cement [16]. This is consistent with the results of our research. In 2013, Thienpont et al. [17] were in line with Niskanen’s results regarding CRP course following TKA and THA. They observed serologic normalization within 21–42 days after the index procedure [17]. Windisch et al. [18] examined the CRP levels following more than 1000 primary TKA and found sex-specific differences in the postoperative trend [18]. Male patients showed significantly higher CRP peaks, in contrast to our present findings. The discrepancy between the sexes was explained by the larger wound area and larger bone resections in male patients due to increased physical size in men. These findings are not supported by the present data considering SA. This appears insofar plausible since CRP defines a parameter relative to body and blood volume [4]. Larsson et al. [9] reported different levels of CRP extent depending on the type of damaged tissue. Bone trauma led to higher CRP levels than fat and soft-tissue damage after TKA and THA [9]. Gender-specific differences were not observed. This is consistent with the findings of Shen et al. [19], who also examined the influence of tissue trauma on the extent of CRP increase [19]. In the present study, increased bone manipulation, as in humeral stem preparation in TSA and RSA, also led to higher absolute peak values in comparison with stemless prostheses or hemiprostheses. However, it must also be kept in mind that patients requiring RSA are generally older, frailer, and more morbid. This could also have an impact on the postoperative course of CRP. Furthermore, not all influences on postoperative CRP levels are known. The individual CRP course should therefore focus on the current trend.

### 4.2. CRP Kinetics Following Shoulder Arthroplasty

Although there is quite a lot of literature published about serum CRP levels following THA and TKA, there is a lack of data dealing with SA [8,20]. One of the few studies investigating the postoperative trends following SA was published by Torrens et al. in 2017 [10]. This prospective study included 58 patients, 41 of whom received RSA. The general CRP trend was similar to the results of the present study, but the CRP peak was mainly observed at the second postoperative day. In contrast, patients who underwent RSA in our cohort predominantly showed the highest serum values at day three. Torrens et al. showed that CRP decreased almost to preoperative baseline on day 14. Torrens et al. conclude that being aware of this potential normal trajectory could be helpful for early detection of acute PJI due to divergent performance of the trend. In our study, even the small number of patients (8%), who showed a delayed peak after the third day, did not differ from the general trend that CRP then decreased consecutively. This fact and the findings of Torrens et al. underline the typical recurring pattern of serum CRP following SA. This may be seen in the context of physiological variance. Zavala et al. [21] reported that the success of revision of SA depends on early intervention [21]. In addition, Coste et al. [11] described good clinical outcomes after PJI in 71% after early revision with irrigation, component change, and antibiotic treatment [11]. This underlines the importance of fast detection of unusual CRP behavior and checking signs of infection within this constellation [21]. In contrast, Topolski et al. [22] examined the correlation of preoperative CRP values with positive intraoperative microbiological specimens in revision SA and reported that the sensitivity of CRP in chronic PJI is very low [22]. Preoperative CRP levels were elevated in only 25% of patients with culture positive findings [22]. Therefore, it is necessary to differentiate between acute and chronic infections. Serum CRP might here be a poor indicator of chronic PJI because a systemic inflammatory response could be absent during infection with low-virulent microorganisms [23]. Since only patients with a bland postoperative course were included in the present study, no statements can be made here about the course in early onset PJI.

### 4.3. Strengths and Limitations

The present study benefits from the large number of patients included, primarily with SA, in a single-center study, a laboratory follow-up period of at least 5–7 days, and a correspondingly large number of individual values in the database. One shortcoming of the study is the retrospective design and the fact that laboratory parameters were not collected after the patients had been discharged from hospital. Another shortcoming is that not every patient had daily laboratory tests. In addition, the exact time interval in hours between tissue trauma and blood sample collection was not standardized. The present study did not examine the influence of different surgeons on the course of CRP. It can be assumed that the individual surgical procedure, in addition to the different prostheses, also has an influence on the course of CRP or the absolute CRP values. This aspect should also be examined in future studies. However, since there is a clear trend in the presented data and these results basically match with the sparse data from the literature, the findings could be consulted in clinical practice.

## 5. Conclusions

There is a serum CRP trend following SA with a peak on day two or three with a subsequent decrease until day seven. RSA and stemmed TSA showed statistically higher CRP peaks than stemless TSA or HA. The CRP trend was not dependent on sex, BMI, operating time, or the application of bone cement. The presented findings may contribute to a better understanding of the postoperative trajectory after shoulder arthroplasty. The results of this retrospective study should be validated by a prospective study design in the future. Furthermore, in the future, it should be investigated whether there are discrepancies in the early postoperative CRP trend in patients with early- or late-onset PJI.

## Figures and Tables

**Figure 1 jcm-09-03893-f001:**
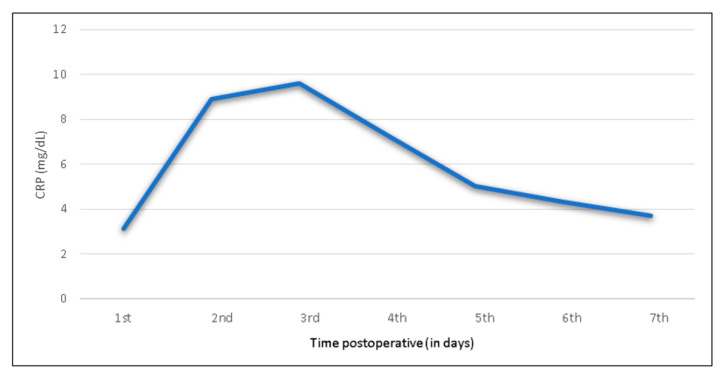
Mean C-reactive protein levels and trend following shoulder arthroplasty in the total study group, illustrating the characteristic trajectory from the first to the seventh postoperative day.

**Figure 2 jcm-09-03893-f002:**
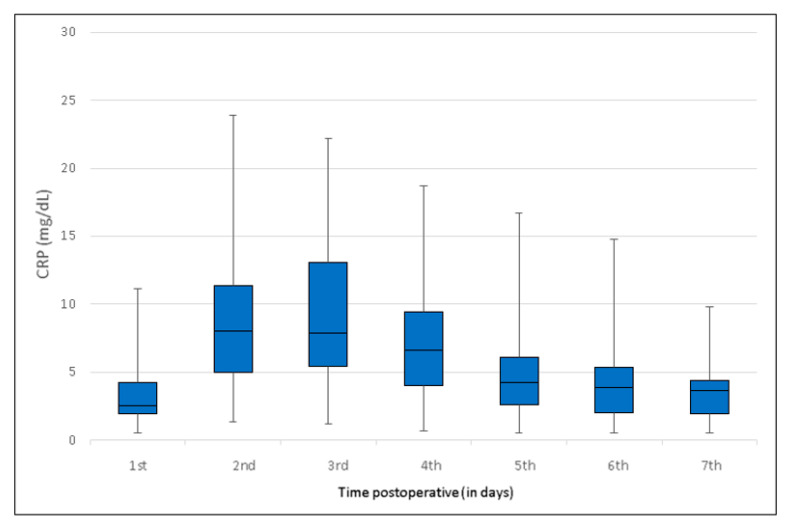
Box-and-whisker plot showing C-reactive protein levels and trend following shoulder arthroplasty in the overall group, illustrating the mean values on days 1 until 7. The blue boxes show the interquartile range (IQR). IQR is the span of the first and third quartiles and is therefore only slightly influenced by outliers. As the IQR refers to the first to third quartiles, a total of 50% of the data are herewith included. The black line indicates the median and the whiskers show the maximum and minimum scattering. Generally, the centrally located median indicates a symmetrical distribution of the values.

**Figure 3 jcm-09-03893-f003:**
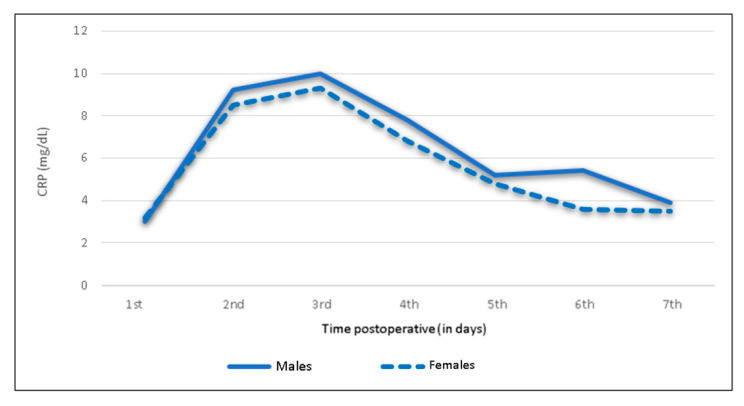
Sex-specific mean C-reactive protein levels and trend following shoulder arthroplasty.

**Figure 4 jcm-09-03893-f004:**
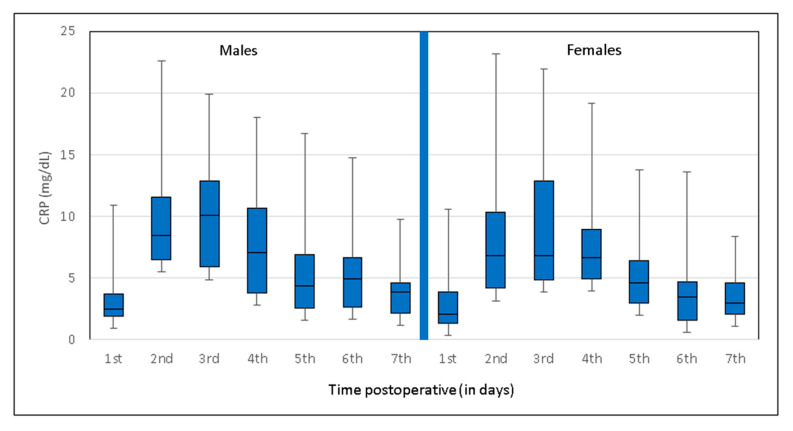
Box-and-whisker plot showing sex-specific C-reactive protein levels and trends following shoulder arthroplasty. Left: Males, right: Females. The blue boxes show the interquartile range (IQR). IQR is the span of the first and third quartiles and is therefore only slightly influenced by outliers. As the IQR refers to the first to third quartiles, a total of 50% of the data are herewith included.

**Figure 5 jcm-09-03893-f005:**
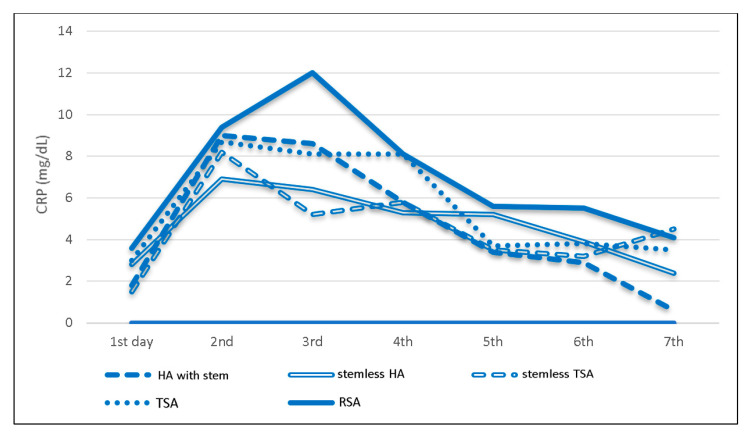
Prosthesis-specific mean C-reactive protein levels and trends following shoulder arthroplasty.

**Figure 6 jcm-09-03893-f006:**
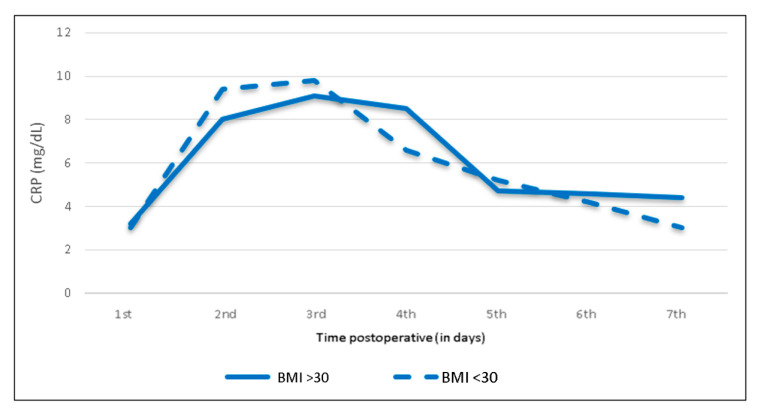
Body-mass-index-specific mean C-reactive protein levels and trends following shoulder arthroplasty. BMI: body mass index.

**Figure 7 jcm-09-03893-f007:**
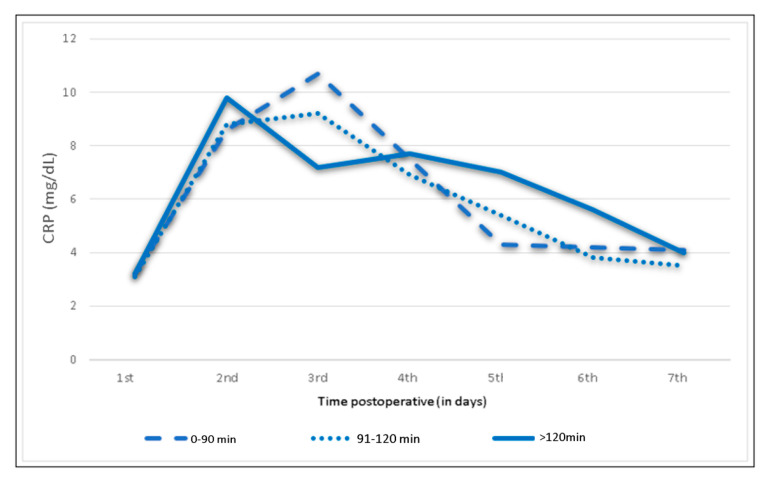
Mean C-reactive protein levels and trends following shoulder arthroplasty depending on the operating time.

**Figure 8 jcm-09-03893-f008:**
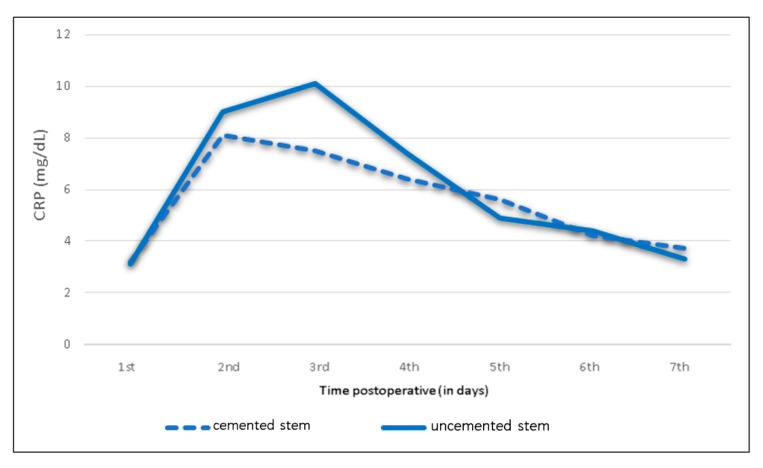
Mean C-reactive protein levels and trends following shoulder arthroplasty depending on application of bone cement.

**Table 1 jcm-09-03893-t001:** Baseline table of the study population presenting the most important demographic findings.

Patient Overview	Total	Male	Female
Patients (*n*)	280	114	166
Age (mean y) (range)	66 (28.1–88.5)	61.2 (28.1–84.1)	69.2 (30.2–88.5)
Hospital stay (mean d) (range)	6.8 (2–14)	6.5 (4–12)	6.9 (2–14)
Hight (mean cm) (range)	169 (148–194)	177 (160–194)	164 (148–180)
Weight (mean kg) (range)	82.6 (40–139)	89.7 (59–130)	77.8 (40–139)
BMI			
(mean kg/m^2^) (range)	28.2 (16–46)	28.2 (19–40)	28.2 (16–46)
<30	176 (63%)	71 (62.3%)	105 (63.3%)
>30	104 (37%)	43 (37.7%)	61 (36.7%)
Side			
Right	155 (55%)	63 (55.3%)	92 (55.4%)
Left	125 (45%)	51 (44.7)	74 (44.6)
Diagnosis			
Primary omarthritis	112 (40%)	48 (42.1%)	64 (38.6%)
Cuff tear arthropathy	99 (35.4%)	35 (30.7%)	64 (38.6%)
Humeral head necrosis	33 (11.8%)	15 (13.1%)	18 (10.8%)
Secondary omarthritis	36 (12.9)	16 (14%)	20 (12%)
Procedure			
Hemicap	2 (0.7%)	2 (1.8%)	0
Stemless	33 (11.8)	14 (12.3%)	19 (11.4%)
TSA	52 (18.6%)	22 (19.3%)	30 (18.1%)
RSA	135 (48.2%)	45 (39.5%)	88 (53.0%)
HA with stem	29 (10.4%)	12 (10.5%)	17 (10.2%)
Stemless HA	29 (10.4%)	19 (16.7%)	10 (6%)
Operating time (min) (range)	102.3 (48–380)	106.5 (60–380)	99.5 (48–186)
Cemented stem			
Yes	52 (18.6%)	15 (13.2%)	37 (22.3%)
No	228 (81.4%)	99 (86.8%)	129 (77.7%)

BMI: body mass index, HA: hemiarthroplasty, RSA: reverse shoulder arthroplasty, TSA: total shoulder arthroplasty.

**Table 2 jcm-09-03893-t002:** Overview of postoperative C-reactive protein levels in mg/dL following shoulder arthroplasty. Mean values and interquartile range in the brackets.

Patients	Preoperative	Day 1	Day 2	Day 3	Day 4	Day 5	Day 6	Day 7
Total group (*n* = 208)	<0.5	3.1 (1.9–4.2)	8.9 (5–11.5)	9.6 (5.3–13.1)	7.3 (3.9–9.7)	5 (2.6–9.8)	4.3 (1.9–5.4)	3.7 (2.0–4.6)
Male (*n* = 114)	<0.5	3 (1.9–3.8)	9.2 (6.2–11.7)	10 (5.7–13.4)	7.8 (3.8–10.8)	5.2 (2.6–7)	5.4 (2.2–6.9)	3.9 (2.2–4.7)
Female (*n* = 166)	<0.5	3.2 (1.9–4.4)	8.5 (4.9–11.2)	9.3 (5.1–13.1)	6.8 (4.1–8.5)	4.8 (2.6–6)	3.6 (1.5–4.7)	3.5 (1.7–4.9)

**Table 3 jcm-09-03893-t003:** Overview of mean postoperative C-reactive protein levels in mg/dL on different days following shoulder arthroplasty according to the different subgroup analysis.

**Patients**	**Preop**	**Day 1**	**Day 2**	**Day 3**	**Day 4**	**Day 5**	**Day 6**	**Day 7**
Total group (*n* = 280)	<0.5	3.1	8.9	9.6	7.3	5.0	4.3	3.7
Male (*n*= 114)	<0.5	3.0	9.2	10.0	7.8	5.2	5.4	3.9
Female (*n*= 166)	<0.5	3.2	8.5	9.3	6.8	4.8	3.6	3.5
Anatomic prostheses	<0.5	2.6	8.0	7.4	6.6	4.0	3.6	3.1
Hemicap (*n* = 2)	<0.5	-	1.3	-	0.9	0.5	0.5	-
HA with stem (*n* = 29)	<0.5	1.8	9.0	8.6	5.8	3.4	2.9	0.6
Stemless HA (*n* = 29)	<0.5	2.8	6.9	6.4	5.3	5.2	3.9	2.4
Stemless TSA (*n* = 33)	<0.5	1.5	8.2	5.2	5.8	3.5	3.2	4.5
TSA (*n* = 52)	<0.5	3.0	8.7	8.1	8.1	3.7	3.8	3.5
RSA (*n* = 135)	<0.5	3.6	9.4	12.0	8.1	5.6	5.5	4.1
BMI > 30 (*n* = 104)	<0.5	3.2	8.0	9.1	8.5	4.7	4.6	4.4
BMI < 30 (*n* = 176)	<0.5	3.0	9.4	9.8	6.6	5.2	4.2	3.0
OP time 0–90 min (*n* = 108)	<0.5	3.2	8.6	10.7	7.5	4.3	4.2	4.1
91–120 min (*n* = 125)	<0.5	3.1	8.8	9.2	6.9	5.4	3.8	3.5
>120 min (*n* = 47)	<0.5	3.2	9.8	7.2	7.7	7.0	5.6	4.0
Cemented stem (*n*= 52)	<0.5	3.2	8.1	7.5	6.4	5.5	4.2	3.7
Uncemented stem (*n*= 228)	<0.5	3.1	9.0	10.1	7.4	4.9	4.4	3.3
**Patients**	**Preop**	**Day 1**	**Day 2**	**Day 3**	**Day 4**	**Day 5**	**Day 6**	**Day 7**
Total group (*n* = 280)	<0.5	3.1	8.9	9.6	7.3	5.0	4.3	3.7
Male (*n* = 114)	<0.5	3.0	9.2	10.0	7.8	5.2	5.4	3.9
Female (*n* = 166)	<0.5	3.2	8.5	9.3	6.8	4.8	3.6	3.5
Anatomic prostheses	<0.5	2.6	8.0	7.4	6.6	4.0	3.6	3.1
Hemicap (*n* = 2)	<0.5	-	1.3	-	0.9	0.5	0.5	-
HA with stem (*n* = 29)	<0.5	1.8	9.0	8.6	5.8	3.4	2.9	0.6
Stemless HA (*n* = 29)	<0.5	2.8	6.9	6.4	5.3	5.2	3.9	2.4
Stemless TSA (*n* = 33)	<0.5	1.5	8.2	5.2	5.8	3.5	3.2	4.5
TSA (*n* = 52)	<0.5	3.0	8.7	8.1	8.1	3.7	3.8	3.5
RSA (*n* = 135)	<0.5	3.6	9.4	12.0	8.1	5.6	5.5	4.1
BMI > 30 (*n* = 104)	<0.5	3.2	8.0	9.1	8.5	4.7	4.6	4.4
BMI < 30 (*n* = 176)	<0.5	3.0	9.4	9.8	6.6	5.2	4.2	3.0
OP time 0–90 min (*n* = 108)	<0.5	3.2	8.6	10.7	7.5	4.3	4.2	4.1
91–120 min (*n* = 125)	<0.5	3.1	8.8	9.2	6.9	5.4	3.8	3.5
>120 min (*n* = 47)	<0.5	3.2	9.8	7.2	7.7	7.0	5.6	4.0
Cemented stem (*n* = 52)	<0.5	3.2	8.1	7.5	6.4	5.5	4.2	3.7
Uncemented stem (*n* = 228)	<0.5	3.1	9.0	10.1	7.4	4.9	4.4	3.3

BMI: body mass index, HA: hemiarthroplasty, OP: operation, RSA: reverse shoulder arthroplasty, TSA: total shoulder arthroplasty.

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
