# Peer review of "Postoperative Trends of Serum C-Reactive Protein Levels after Primary Shoulder Arthroplasty—Normal Trajectory and Influencing Factors"

_jcm, 2020, doi:10.3390/jcm9123893_

Round 1

Reviewer 1 Report

Dear sir, the work presents a large population study but in a retrospective way. The results should be written in a simpler way . I belive that PCR curve in reverse prosthesis should be discused (the size of implant etc). The comment about infection is very long and the work does nor address that issue.

Author Response

Sincerely,

Sebastian Klingebiel

Reviewer 2 Report

It is indeed well known that a downward trend of CRP values in hip/knee joint arthroplasties would suggest an uneventful recovery from major surgery. Again - trending is the key and one has to be careful with absolute and cut-off values.In this respect, this research confirms that the postop course after major shoulder surgery is no exception.

The authors state that higher post-op CRP values correlate with the type of prosthesis used i.e. read with the magnitude of the actual procedure. I am wondering though whether this observation is based on a preceding exercise by the authors in properly matching the cohorts of prosthetic type to the health status. Perhaps patients requiring a a (reverse) total shoulder arthroplasty did suffer from more or less ailments or frailty  at an advanced age.

Patients remained remarkably long in hospital in a way allowing a very thorough and meticulous serologic follow up. In light of modern daycare or one night stay trends the authors have to be commended for their careful observations of the postoperative course after advanced shoulder surgery. In a way, it would have been useful not to exclude patients who did develop complications within 3 months after the index procedure but rather use that cohort to ascertain that higher post-op CRP values in uncomplicated (reverse) TSR cases can be differentiated from values in  (reverse) TSR or other cases with complications such as SSI or early onset PJI. Perhaps, the authors have already conducted a study in this respect.

The authors particularly excluded patients who developed postoperative complications within the period of up to the postop check of 3 months after the index procedure. It is then rather obvious that cfr. line 221 no patient reported complications.

In conclusion, the physiologic response after major shoulder joint surgery appears to follow similar patterns as in major hip and knee joint procedures.In light of the fact the shoulder procedures in this paper were carried out by a single surgeon, one implies that tissue and bone handling has remained consistent over the decade 2010-2019. The author's statement that absolute CRP values might correlate with type of prothesis used could be considered contentious i.e. a 'rough handling' surgeon might perhaps end up with similar CRP values for his/her less invasive shoulder procedures.

Author Response

Sincerely,

Sebastian Klingebiel
